# Highly Efficient Extracellular Production of Recombinant *Streptomyces* PMF Phospholipase D in *Escherichia coli*

**Jing Wang [†], Sheng Xu [†], Yang Pang, Xin Wang *, Kequan Chen and Pingkai Ouyang**

State Key Laboratory of Materials-Oriented Chemical Engineering, College of Biotechnology and Pharmaceutical Engineering, Nanjing Tech University, Nanjing 211816, China; njtechwj@163.com (J.W.); henryxu@njtech.edu.cn (S.X.); njtechpy@163.com (Y.P.); kqchen@njtech.edu.cn (K.C.); ouyangpk@njtech.edu.cn (P.O.)

* Correspondence: xinwang1988@njtech.edu.cn
† These authors contributed equally to this work.

**Abstract:** To achieve efficient bio-production of phospholipase D (PLD), PLDs from different organisms were expressed in *E. coli*. An efficient secretory expression system was thereby developed for PLD. First, PLDs from *Streptomyces* PMF and *Streptomyces racemochromogenes* were separately over-expressed in *E. coli* to compare their transphosphatidylation activity based on the synthesis of phosphatidylserine (PS), and PLD$_{PMF}$ was determined to have higher activity. To further improve PLD$_{PMF}$ synthesis, a secretory expression system suitable for PLD$_{PMF}$ was constructed and optimized with different signal peptides. The highest secretory efficiency was observed when the PLD * (PLD$_{PMF}$ with the native signal peptide Nat removed) was expressed fused with the fusion signal peptide PelB-Nat in *E. coli*. The fermentation conditions were also investigated to increase the production of recombinant PLD and 10.5 U/mL PLD was ultimately obtained under the optimized conditions. For the application of recombinant PLD to PS synthesis, the PLD properties were characterized and 30.2 g/L of PS was produced after 24 h of bioconversion when 50 g/L phosphatidylcholine (PC) was added.

**Keywords:** phospholipase D; phosphatidylserine; secretory expression; enzymatic catalysis

## 1. Introduction

Phospholipase D (PLD, EC 3.1.4.4) catalyzes hydrolysis of the phosphodiester bond of glycerophospholipids to generate phosphatidic acid and a free head-group. In addition to its hydrolytic activity, PLD can also catalyze the transfer of acyl groups to directly synthesize valuable phospholipid derivatives, such as phosphatidylethanolamine (PE), phosphatidylserine (PS) and phosphatidylglycerol (PG) [1]. These phospholipids have wide applications in the food, cosmetics and pharmaceutical industries [2]. PLD was first reported in 1947 and due to its special catalytic activity, research on PLD has recently increased [3]. PLD has been identified from plants [4], mammals [5] and bacteria [6]. However, these natural sources produce low levels of PLD that cannot meet the industrial demand [7]. Therefore, the production of PLD by microbial fermentation has attracted great attention due to its advantages of high unit activity and low cost.

PLD has been characterized in many microorganisms and is most commonly found in *Streptomyces* strains, such as *Streptomyces* PMF [8], *S. lividans* [9], *S. racemochromogenes* [10] and *Streptomyces* sp. YU100 [11]. Compared with PLD from *Streptomyces*, PLD coming from other organisms has the transphosphatidylation activity, the activity is much lower [12]. For the production of PLD which has great potential in the industrial synthesis of high-value-added phospholipids, *Streptomyces* strains are most widely used due to the high transphosphatidylation activity of native PLD. For example,

Saovanee et al. isolated *Streptomyces* sp. SC734 from soil-contaminated palm oil, and the PLD it produced exhibited high activity with a conversion rate of phosphatidylcholine (PC) to PS of up to 94.7% in 100 min [13]. Ogino et al. constructed an overexpression system for secretory production of PLD in *S. lividans* and the amount of PLD secreted reached a maximum level of 118 mg/L [14]. However, the genetic transfer systems for *Streptomyces* remain largely inefficient, which limits efficient production of PLD. Thus, the heterologous expression of *Streptomyces* PLDs in other model microorganisms, such as yeast or *E. coli* is highly desired.

Using *Pichia pastoris* as the host, Liu et al. developed a yeast cell surface display system to express PLD from *S. chromofuscus,* and the displayed PLD converted 67.5% of PC to PS within 10 h [15]. PLDs from different sources have also been successfully expressed in *E. coli.* For example, Zambonelli et al. expressed the PLD from *Streptomyces* PMF in *E. coli* BL21(DE3)pLysS, and 5 mg/L PLD was finally obtained with an enzyme activity of 15 mU/mL [16]. For the high-level and stable production of PLD, several engineering strategies were carried out in *E. coli*, including optimizing and tightly regulating promoter strength, optimizing codon usage and amino acid supplementation, and maintaining the best cellular state by supplementing nutrition. Finally, a large amount of PLD (81.5 mg/L) was obtained in batch culture [17]. Although there has been considerable progress in heterologous production of PLD, it is still not enough for industrial applications of PLD. Developing an efficient expression system for PLD production is urgently needed.

One of the biggest obstacles to efficient PLD production is that overexpressed PLD is toxic to the host, which may cause plasmid instability, cell lysis and PLD leakage [17]. Secretory production of heterologous proteins has great advantages compared with conventional cytosolic protein production, especially when the heterologous proteins are toxic. In addition, the secretory production of heterologous proteins could simplify the purification processes and reduce cost since cell disruption is not required. Many reports have proposed strategies for improving the secretory production of heterologous proteins, such as co-expression of the signal peptide [18], optimizing the environmental conditions [19], constructing leaky strains [20] or co-expressing the secretory pathway [21]. The production of PLD in the secretory form seems to be a promising approach to address this issue.

In this study, PLDs from *Streptomyces* PMF and *Streptomyces racemochromogenes* were separately overexpressed in *E. coli* to compare their transphosphatidylation activity based on synthesis of PS. Recombinant $PLD_{PMF}$ exhibited higher activity. To further improve the synthesis of $PLD_{PMF}$, a secretory expression system suitable for $PLD_{PMF}$ was constructed and optimized with different signal peptides. The highest secretory efficiency was observed when the PLD * was expressed fused with the fusion signal peptide Nat-PelB. After optimizing induction conditions including induction temperature, induction pH, IPTG concentration, induction time and addition of metal ions, 10.5 U/mL PLD was detected in the fermentation medium. For the application of recombinant PLD to PS synthesis, the PLD properties were characterized and 30.2 g/L of PS was produced after bioconversion for 24 h when 50 g/L PC was added.

## 2. Results

### 2.1. Intracellular Expression of PLD in E. coli

The host strain *E. coli* BL21(DE3) is an efficient expression system for various recombinant proteins. Here we attempted to use it for the production of PLD. Two PLDs in the plasmids pET28a-$PLD_{PMF}$ and pET28a-$PLD_{SR}$ were separately introduced into BL21(DE3). Enzyme production was induced by the addition of IPTG and the activities of crude PLD extracts were compared. As shown in Figure 1, both PLDs were functionally expressed in *E. coli*, and crude extracts of the strain BL21(DE3)/pET28a-$PLD_{PMF}$ exhibited higher transphosphatidylation activity. Using the intracellular fraction of the strain BL21(DE3)/pET28a-$PLD_{PMF}$ for the bioconversion of PC to PS, PS reached 0.37 g/L after 8 h, which is 1.4-fold higher than that of BL21(DE3)/pET28a-$PLD_{SR}$. Besides this, we also tested

the catalytic activity of the extracellular fraction of these two strains. Unfortunately, there was no PS detected. This result suggested that, almost all heterologous produced PLD was in the cell and the signal peptide from *Streptomyces* could not guide the secretion of PLD when it was expressed in *E. coli*. Relatively speaking, $PLD_{PMF}$ may be more suitable for expression in *E. coli*, although this may be a comprehensive result caused by factors such as intrinsic enzyme catalytic activity and enzyme production. Therefore, $PLD_{PMF}$ was applied to further optimize expression.

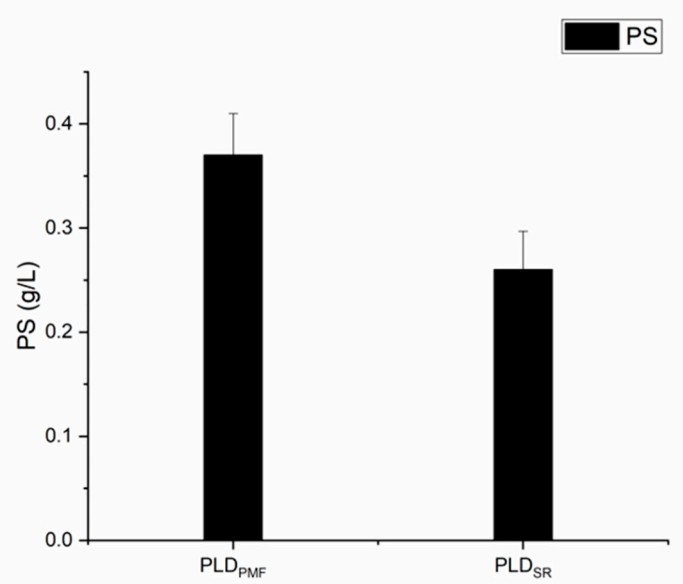

**Figure 1.** Comparison of the transphosphatidylation activity of the phospholipase D (PLD) coming from *Streptomyces* PMF and *S. racemochromogenes* when expressed in *E. coli*.

### 2.2. Secretory Expression of PLD in E. coli by Optimizing Signal Peptides

To investigate the secretory expression of PLD, the signal peptides Nat (Native signal peptide from $PLD_{PMF}$), OmpA and the fusion signal peptide OmpA-Nat were fused-expressed with the PLD * (Figure 2a). First, the effect of Nat and OmpA on the PLD secretory efficiency was compared. No PS was detected using the extracellular fraction of the strain BL21(DE3)/pET22b-Nat-PLD * for the conversion of PC to PS (Figure 2b), this result proved that signal peptide Nat could not guide the secretion of PLD when it was expressed in *E. coli* again. In contrast, the PS yield reached 40.68% after bioconversion for 24 h using the extracellular fraction of the strain BL21(DE3)/pET22b-OmpA-PLD *, indicating that OmpA is functional for directing the secretion of heterologous PLD in *E. coli*. Subsequently, to identify whether the cleavage of Nat sequence in the N-terminus of $PLD_{PMF}$ affected PLD activity, PLD * was expressed after being fused with OmpA-Nat in *E. coli*. Compared with PLD produced by BL21 (DE3)/pET22b-OmpA-PLD *, We found that the PLD from the strain BL21(DE3)/pET22b-OmpA-Nat-PLD * exhibited higher transphosphatidylation activity with PS yield of 45.72% after bioconversion for 24 h. This result indicated that the Nat signal peptide may be important for maintaining the transphosphatidylation activity of recombinant PLD. Thus, the fused signal peptide was more suitable for the secretory expression of PLD in *E. coli*.

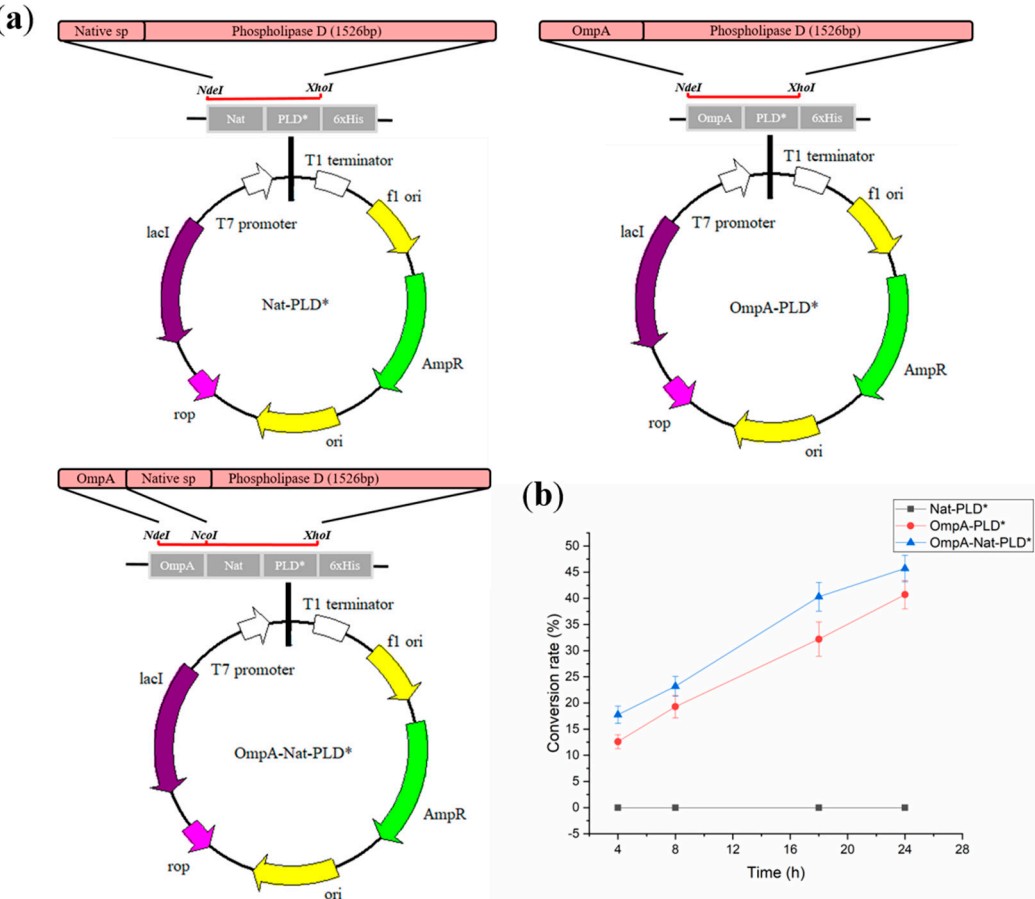

**Figure 2.** Effects of the Nat signal peptide on the transphosphatidyltion activity of recombinant PLD$_{PMF}$ expressed by *E. coli*. (**a**) Structure of the plasmids Nat-PLD *, OmpA-PLD * and OmpA-Nat-PLD *. (**b**) The conversion rates of phosphatidylcholine (PC) to phosphatidylserine (PS) catalyzed by the extracellular PLD expressed by the strain: BL21(DE3)/Nat-PLD *, BL21(DE3)/OmpA-PLD * and BL21(DE3)/OmpA-Nat-PLD *.

To further determine the optimum signal peptide for directing secretion expression of PLD in *E. coli*, seven different signal peptides were employed to replace OmpA to form new fusion signal peptides. (Figure 3a). As shown in Figure 3b, different signal peptides directed export of PLD with varying efficiencies. Compared with OmpA-Nat, the fused signal peptides OmpF-Nat, OmpT-Nat, LamB-Nat and MalE-Nat are less efficient and lower PS yield was observed. In contrast, more efficient PLD secretion was obtained with the signal peptides OmpC-Nat, PhoA-Nat and PelB-Nat resulting in higher PS yield. Among them, the highest level of extracellular PLD * was found after expressing the plasmid PelB-Nat-PLD * in *E. coli*, where PS yield was increased by 86.51% compared to that from OmpA-Nat. Thus, the recombinant strain BL21(DE3)/pET22b-PelB-Nat-PLD * with the highest PLD secretory expression activity was selected for the following experiment.

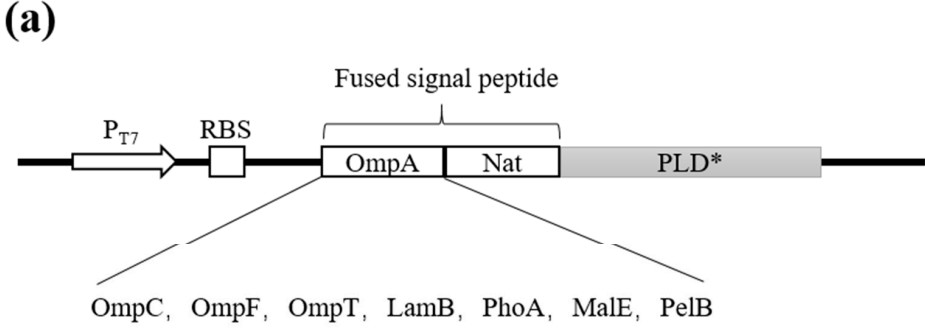

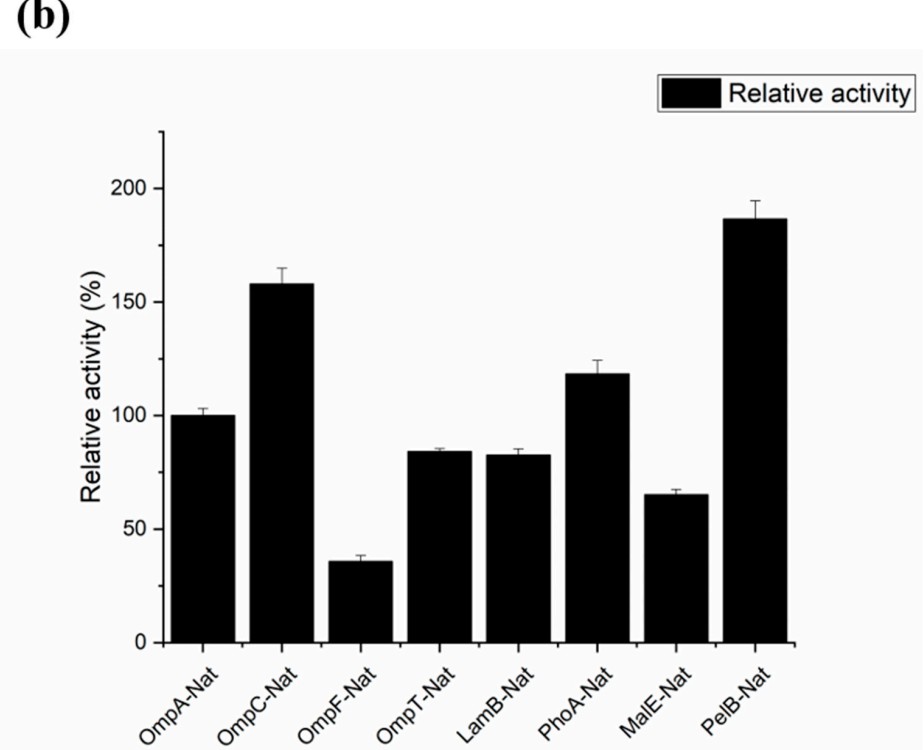

**Figure 3.** Optimization of the signal peptides for the secretory expression of PLD * in *E. coli*. (**a**) Using different signal peptides from *E. coli* to replace OmpA to construct different fused signal peptides. (**b**) The relative transphosphatidylation activity of recombinant PLD * when fused expressed in *E. coli* with different fusion signal peptides. The activity of PLD expressed by the strain BL21(DE3)/pET22b-OmpA-Nat-PLD * was the reference (100% relative activity).

## 2.3. Effect of Fermentation Conditions on the Secretory Expression of PLD

To further improve PLD synthesis, fermentation conditions, including induction temperature, induction pH, cell density at induction and IPTG concentration were optimized. For the control group, the induction temperature was 28 °C, IPTG concentration was 0.5 mm, original cultivation pH was 7.0, no surfactant was added, induction time was 12 h and the induction $OD_{600nm}$ was 0.6. Compared with the control group, each group of experiments has only one single variable. Induction temperature is an important factor influencing heterologous protein expression in *E. coli*. The recombinant strain was incubated at a temperature ranging from 16 °C to 36 °C, and the maximum PLD activity was obtained at 20 °C with an increase of 19.4% compared to the control group (Figure 4a). The effect of the concentration of IPTG was evaluated by varying the concentration from 0.4 mm to 0.8 mm. The highest PLD activity was achieved when 0.7 mm IPTG was added, which resulted in a 68.2% increase in PS yield (Figure 4b). The optimal induction $OD_{600nm}$ and time were also determined at the induction

$OD_{600}$ of 1.4 after induction for 12 h (Figure 4c,d). Varying pH of the growth environment change bacterial metabolic pathways, which might negatively affect the expression of heterologous protein in *E. coli*. In addition, pH also affects the charge state on the cell surface, and thus the permeability of the cell membrane, which has an important impact on the exchange of substances and the secretion of recombinant proteins. When the engineered BL21(DE3)/pET22b-PelB-Nat-PLD * was cultivated at pH ranging from 5.0 to 8.0, the most suitable original cultivation pH for the secretory expression of PLD was 6.5 (Figure 4e).

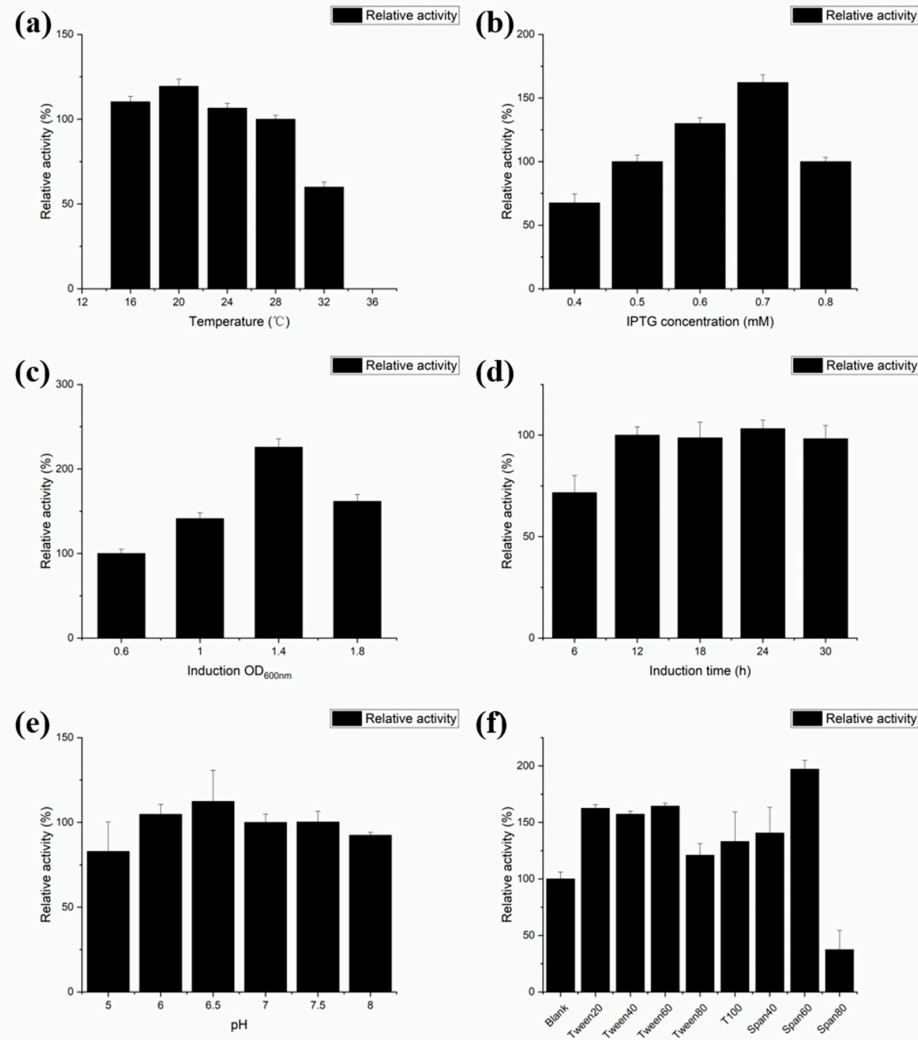

**Figure 4.** Optimization of the fermentation conditions for the secretory expression of $PLD_{PMF}$. (**a**) The effect of induction temperature. (**b**) The effect of the concentration of β-d-1-thiogalactopyranoside (IPTG). (**c**) The effect of the induction $OD_{600nm}$. (**d**) The effect of the induction time. (**e**) The effect of the original cultivation pH. (**f**) The effect of the addition of surfactants.

To further improve the secretory expression of recombinant PLD, the effects of surfactant addition were evaluated. As shown in Figure 4f, seven different surfactants were separately added into the fermentation medium. Compared to the control group, all surfactants benefitted the secretory expression of PLD, among which, the group with 3 g/L Span60 added exhibited the best PLD activity.

## 2.4. Characterization of the Recombinant PLD Activity

To characterize the recombinant PLD activity, the effects of reaction temperature, pH and metal ion additives were evaluated. The initial reaction was carried out under the following conditions:

30 °C, pH 5.5, no metal ions addition, and the activity of PLD was used as the reference value (100% relative activity). To determine the optimal reaction temperature, bioconversion was carried out at 20, 25, 30, 35 or 40 °C (Figure 5a). From 20 °C and 30 °C, the PLD activity clearly increased with increasing temperature, and reached the highest level at 30 °C. When the temperature was higher than 30 °C, the PLD activity sharply decreased, indicating the temperature sensitivity of recombinant PLD. The PLD activity increased with a rise in reaction pH (from 4.0 to 8.0) and reached a maximum at pH 5.5 (Figure 5b).

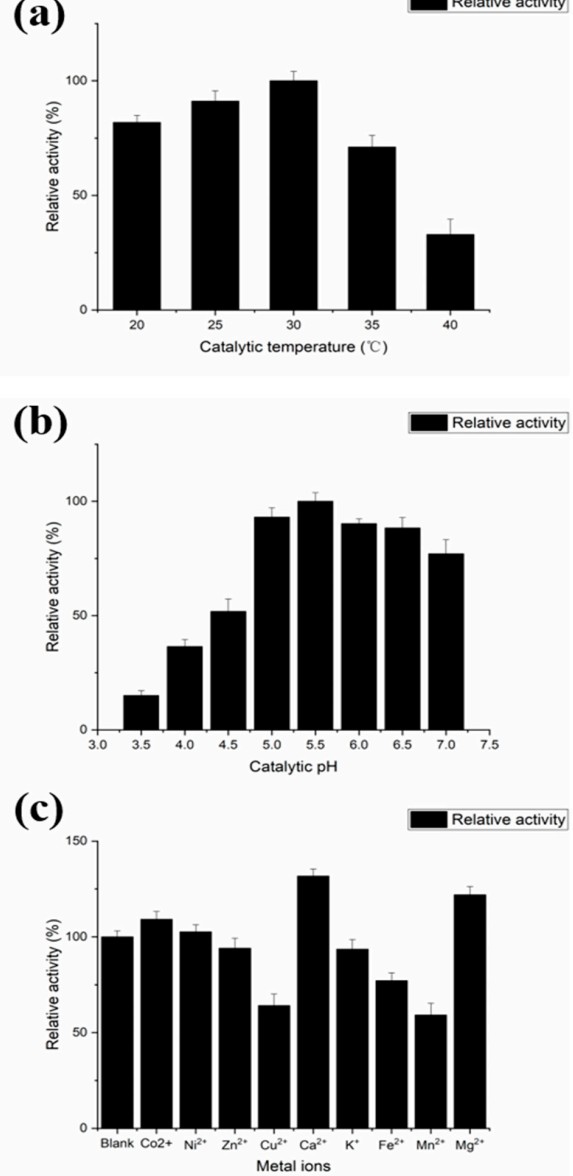

**Figure 5.** Characterization of the recombinant $PLD_{PMF}$. (**a**) The effect of the catalytic temperature. (**b**) The effect of the catalytic pH. (**c**) The effect of the addition of metal ions.

Several metal ions have been reported to play an important role in maintaining the activity of PLD [22]. To evaluate the effect of metal ions on recombinant PLD activity, 10 mm metal ion ($Co^{2+}$, $Ni^{2+}$, $Zn^{2+}$, $Cu^{2+}$, $Ca^{2+}$, $K^+$, $Fe^{2+}$, $Mg^{2+}$, $Mn^{2+}$) was added into the aqueous phase. The reaction performed without any metal ions served as the control group. The addition of $Co^{2+}$, $Ca^{2+}$ and $Mg^{2+}$ showed positive effects on the PLD activity, and the addition of $Ca^{2+}$ gave the highest level of PLD activity (Figure 5c).

### 2.5. The Application of Recombinant PLD$_{PMF}$ for the Bioconversion of PC to PS

The optimum expression system (PelB-Nat fused-expressed with PLD *), fermentation conditions (induction temperature: 20 °C; IPTG concentration: 0.7 mm; induction OD$_{600nm}$: 1.4; induction time: 12 h; original fermentation pH: 6.5; addition of surfactant: 3 g/L Span60) and PLD traits (catalytic temperature: 30 °C; original catalytic pH: 5.5; addition of metal ions: 10 mm Ca$^{2+}$) were determined based on the above results. With the extracellular recombinant PLD produced by engineered *E. coli*, the capacity of producing PS from PC by PLD was tested under the optimal reaction conditions. The reaction was performed with PC substrate concentrations of 10 g/L, 30 g/L and 50 g/L. Samples were taken at reaction times of 4 h, 8 h, 12 h and 24 h to detect the amount of product PS and substrate PC (Figure 6). In the case of 10 g/L PC, after bioconversion of 24 h, the production of PS reached 9.2 g/L with a molar yield of 88.05%. When the concentration of PC was increased to 30 g/L, the final PS titer of 18.2 g/L was obtained with a molar yield of 58.02% after bioconversion of 24 h. Further increasing PC concentration to 50 g/L, increased PS titer to 30.2 g/L with a molar yield of 57.81%.

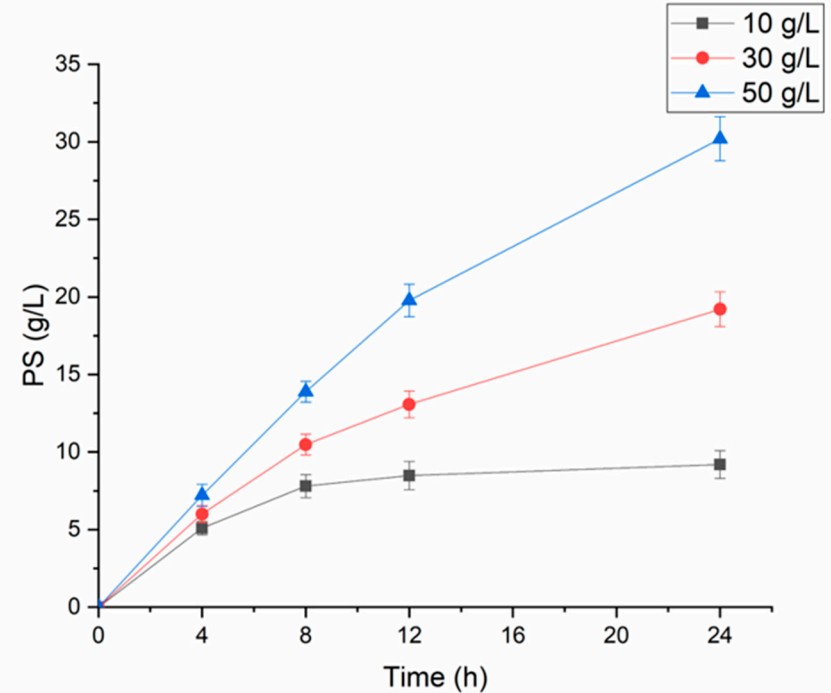

**Figure 6.** Using the recombinant PLD$_{PMF}$ for the bioconversion of PC to PS.

## 3. Discussion

Phospholipase has achieved significant attention in recent years for its applications in the production of various high value rare phospholipids [23]. To achieve high production of PLD, overexpression of native PLD or heterologous expression of various PLDs in model microorganisms, including *E. coli* [24], yeast [25] and *Bacillus subtilis* [26] have been performed. Among them, *E. coli* is the most frequently used host strain for the expression of heterologous proteins due to its well-characterized genetics, high protein expression levels and rapid growth rate [27]. However, the toxicity from overexpressing PLD has limited its production in *E. coli*. In this work, after screening PLD sources, a secretory PLD expression system was developed and optimized by investigating the effects of different signal peptides for efficient PLD production.

After determining a suitable source of PLD, OmpA, a signal peptide that has been reported to guide the secretory expression of heterologous proteins with high efficiency in *E. coli* [28], was first employed to direct the secretory production of recombinant PLD. The effect of native signal peptide (Nat) in PLD$_{PMF}$ sequence was also evaluated. Fortunately, the PLD production efficiency was largely enhanced

with the secretory expression system. Moreover, the presence of Nat signal peptides resulted in a higher extracellular PLD activity, suggesting that the Nat sequence might contribute to the correct fold of recombinant PLD in *E. coli*. It is well known that the N-terminal signal sequence can guide the protein to the Sec-translocon through the post-translational SecB-targeting pathway or the co-translational signal recognition particle (SRP)-targeting pathway and then fold correctly [29]. However, there is currently no general rule in selecting a proper signal sequence for a given recombinant protein [30]. To identify a more efficient signal peptide to direct PLD secretion in *E. coli,* seven other signal peptides including OmpF-Nat, OmpT-Nat, OmpC-Nat, LamB-Nat, MalE-Nat, PhoA-Nat and PelB-Nat were compared. The highest secreted PLD activity occurred with PelB-Nat (Figure 3b), indicating the important role of signal peptide for the efficient production of recombinant PLD.

For the secretory production of recombinant proteins, membrane permeability might be a limiting factor since the cellular membrane often retards the entry of substrate into the cellular systems and prevents the product from being released from the cellular system for an easy recovery [31]. With the addition of 3 g/L Span60, the extracellular PLD activity was increased by 97.1%, indicating that cell membrane permeability is one of the key factors affecting secretory expression of recombinant PLD in *E. coli*. To address this issue, co-expression of the key secretion components, construction of leaky strains and utilization of different secretion pathways to enhance secretory production of heterologous PLD could be further carried out in future studies.

For applying PLD to the synthesis of high-value-added phospholipid, the properties of recombinant PLD were also characterized. As shown in Figure 5a, recombinant PLD is sensitive to temperature changes. The best pH for PLD activity was observed under pH 5.5, which coincides with other reports in which PLD exerted high transphosphatidylation activity in a weak acid environment [32]. For the metal ion additives, the highest PLD activity was observed with the addition of $Ca^{2+}$ [12]. $Ca^{2+}$ binding to PLD has been reported to cause a conformational change in the PLD that enhances binding of protein to zwitterionic interfaces [13]. $Ca^{2+}$ is also an activator when other soluble substrates are used [33]. $Ca^{2+}$ possibly coordinates with enzymes, improving their stability. Finally, the recombinant PLD was applied for the bioconversion of PC to PS.

Previous reports on the enzymatic synthesis of PS focused on the use of PLDs expressed in *Streptomyces*. PS synthesis with a conversion rate of 88% was documented using the PLD from *S. racemochromogenes* [10]. Duan and Hu compared five commercial PLDs in the synthesis of PS, and PLD derived from *S. chromofuscus* achieved 90% yield of PS after 12 h of bioconversion [34]. In our work, the recombinant PLD converted 88.05% of PC into PS with a concentration of 10 g/L PC, indicating the high transphosphatidylation activity of the recombinant PLD expressed by *E. coli*. Under optimized conditions, 30.2 g/L PS was obtained with a yield of 57.81%. The recombinant PLDs obtained in *E. coli* are summarized in Table 1, and the highest PS concentration so far was obtained in our study.

**Table 1.** Production of recombinant PLD in *E. coli* and synthesis of PS.

| PLD Origin | Expression Host | PS (g/L) | References |
|---|---|---|---|
| *Streptomyces mobaraensis* | *E. coli* | 0.2 | [12] |
| *Streptomyces chromofuscus* | *E. coli* | 3.94 | [35] |
| *Streptomyces* sp. YU100 | *E. coli* | ND | [11] |
| *Streptomyces antibioticus* | *E. coli* | ND | [17] |
| *Streptomyces* PMF | *E. coli* | 30.2 | This study |

## 4. Materials and Methods

### 4.1. Microorganisms and Media

The strains used and constructed in this paper are listed in Table 2. The *E. coli* strains were cultured in Luria–Bertani medium (tryptone 10 g/L, NaCl 5 g/L and yeast extract 5 g/L) containing appropriate antibiotics at the following concentrations: 50 mg/L kanamycin (kana) and 100 mg/L

ampicillin (Amp). Plasmid pET28a was used as the original plasmid. L-serine, phosphatidylserine (PS) and phosphatidylcholine (PC) were purchased from Aladdin Ind. Co., Ltd. (China). $CoCl_2 \cdot 6\ H_2O$, KCl, $CaCl_2$, $MgCl_2 \cdot 6\ H_2O$, $FeCl_2$, $MnCl_2 \cdot 4\ H_2O$, $NiCl_2 \cdot 6\ H_2O$ and $ZnCl_2$ were purchased from Xilong Chemical Company (China).

**Table 2.** Strains and plasmids used in this study.

| Strains or Plasmids | Characteristics | Sources |
|---|---|---|
| **Strains** | - | - |
| *Streptomyces* PMF | Source of $PLD_{PMF}$ gene | ATCC |
| *Streptomyces racemochromogenes* | Source of $PLD_{SR}$ gene | ATCC |
| *E. coli* DH5$\alpha$ | Used as cloning vector | Invitrogen |
| *E. coli* BL21(DE3) | Used as expression host | Invitrogen |
| BL21(DE3)/pET28a-$PLD_{PMF}$ | Express plasmid: pET28a-$PLD_{PMF}$ | This study |
| BL21(DE3)/pET28a-$PLD_{SR}$ | Express plasmid: pET28a-$PLD_{SR}$ | This study |
| BL21(DE3)/pET22b-$PLD_{PMF}$ | Express plasmid: pET22b-$PLD_{PMF}$ | This study |
| BL21(DE3)/Nat-PLD * | Express plasmid: Nat-PLD * | This study |
| BL21(DE3)/OmpA-PLD * | Express plasmid: OmpA-PLD * | This study |
| BL21(DE3)/OmpA-Nat-PLD * | Express plasmid: OmpA-Nat-PLD * | This study |
| BL21(DE3)/OmpC-Nat-PLD * | Express plasmid: OmpC-Nat-PLD * | This study |
| BL21(DE3)/OmpF-Nat-PLD * | Express plasmid: OmpF-Nat-PLD * | This study |
| BL21(DE3)/OmpT-Nat-PLD * | Express plasmid: OmpT-Nat-PLD * | This study |
| BL21(DE3)/LamB-Nat-PLD * | Express plasmid: LamB-Nat-PLD * | This study |
| BL21(DE3)/PhoA-Nat-PLD * | Express plasmid: PhoA-Nat-PLD * | This study |
| BL21(DE3)/MalE-Nat-PLD * | Express plasmid: MalE-Nat-PLD * | This study |
| BL21(DE3)/PelB-Nat-PLD * | Express plasmid: PelB-Nat-PLD * | This study |
| **Plasmids** | - | - |
| pET28a | pBR322 ori, $P_{T7}$, $Kan^R$ | Our lab |
| pET22b | pBR322 ori, $P_{T7}$, OmpA signal peptide, $Amp^R$ | Our lab |
| pET28a-$PLD_{PMF}$ | pET28a derivative; $P_{T7\text{-lacO}}$-$PLD_{PMF}$ | This study |
| pET28a-$PLD_{SR}$ | pET28a derivative; $P_{T7\text{-lacO}}$-$PLD_{SR}$ | This study |
| pET22b-$PLD_{PMF}$ | pET22b derivative; $P_{T7\text{-lacO}}$-OmpA-$PLD_{PMF}$ | This study |
| Nat-PLD * | pET22b derivative; $P_{T7\text{-lacO}}$-Nat-PLD * | This study |
| OmpA-PLD * | pET22b derivative; $P_{T7\text{-lacO}}$-OmpA-PLD * | This study |
| OmpA-Nat-PLD * | pET22b derivative; $P_{T7\text{-lacO}}$-OmpA-Nat-PLD * | This study |
| OmpC-Nat-PLD * | pET22b derivative; $P_{T7\text{-lacO}}$-OmpC-Nat-PLD * | This study |
| OmpF-Nat-PLD * | pET22b derivative; $P_{T7\text{-lacO}}$-OmpF-Nat-PLD * | This study |
| OmpT-Nat-PLD * | pET22b derivative; $P_{T7\text{-lacO}}$-OmpT-Nat-PLD * | This study |
| LamB-Nat-PLD * | pET22b derivative; $P_{T7\text{-lacO}}$-LamB-Nat-PLD * | This study |
| PhoA-Nat-PLD * | pET22b derivative; $P_{T7\text{-lacO}}$-PhoA-Nat-PLD * | This study |
| MalE-Nat-PLD * | pET22b derivative; $P_{T7\text{-lacO}}$-MalE-Nat-PLD * | This study |
| PelB-Nat-PLD * | pET22b derivative; $P_{T7\text{-lacO}}$-PelB-Nat-PLD * | This study |

PLD *: the $PLD_{PMF}$ with the native signal peptide (Nat) removed.

### 4.2. Plasmid Construction

All the primers used in this study are listed in Table 3. Two PLD gene fragments, $PLD_{PMF}$ and $PLD_{SR}$ were amplified from genomic DNA of *Streptomyces* PMF and *Streptomyces racemochromogenes*, respectively. The signal peptide genes OmpC, OmpF, OmpT, LamB, PhoA, MalE and pelB were synthesized by Sprin GenBioTech Co. LTD (Nanjing, China). The codon optimization procedure for the two PLD genes were conducted by Sprin GenBioTech Co. LTD (Nanjing, China). We used the primers P1 and P2 to amplify the $PLD_{PMF}$ gene and inserted it into the plasmid pET28a between the *Nco*I and *Eco*RI sites, yielding the recombinant plasmid pET28a-$PLD_{PMF}$. The $PLD_{SR}$ fragment was amplified using the primer P3 with a *Nco*I restriction site and primer P4 with an *Eco*RI restriction site, and was ligated into pET-28a vector, yielding the plasmid pET28a-$PLD_{SR}$.

**Table 3.** Primers used in this study.

| Name | Primers | Sequences (5′–3′) |
|------|---------|-------------------|
| P1 | NcoI-PLD$_{PMF}$-F | CATGCCATGGCAGCTGACTCTGCTACCCCG |
| P2 | EcoRI-PLD$_{PMF}$-R | CCGGAATTCTCAGGCGTTGCAGATCCC |
| P3 | NcoI-PLD$_{SR}$-F | CATGCCATGGGTGCGGAGGTGTGGTCGTAC |
| P4 | EcoRI-PLD$_{SR}$-R | CCGGAATTCTCAGGCCTGGCAGAGG |
| P5 | NdeI-Nat-PLD * | GGAATTCCATATGCTACATGGGTCACACCT |
| P6 | XhoI-Nat-PLD * | CTCGAGCGGAGCGTTGCAGATACCAC |
| P7 | NcoI-OmpA-Nat-PLD * | CCATGGGCTACATGGGTCACA |
| P8 | XhoI-OmpA-Nat-PLD * | CTCGAGCGGAGCGTTGCAGATACCAC |
| P9 | NcoI-OmpA-PLD * | CATGCCATGGCAGCTGACTCTGCTACCCCG |
| P10 | XhoI-OmpA-PLD * | CTCGAGCGGAGCGTTGCAGATACCAC |
| P11 | LamB-F | GGAATTCCATATGATTACTCTGCGCAAACTTCCTCTGGCGGTTGCCGTCGCAGCGGGCGTAATGTCTGCTCAGGCAATGGCTCCATGGGCTACATGGGTCACA |
| P12 | MalE-F | GGAATTCCATATGAAAATAAAAACAGGTGCACGCATCCTCGCATTATCCGCATTAACGACGATGATGTTTTCCGCCTCGGCTCTCGCCCCATGGGCTACATGGGTCACA |
| P13 | OmpC-F | GGAATTCCATATGAAAGTTAAAGTACTGTCCCTCCTGGTCCCAGCTCTGCTGGTAGCAGGCGGCAGCAAACGCTCCATGGGCTACATGGGTCACA |
| P14 | OmpF-F | GGAATTCCATATGAAGCGCAATATTCTGGCAGTGATCGTCCCTGCTCTGTTAGTAGCAGGTACTGCAAACGCTCCATGGGCTACATGGGTCACA |
| P15 | OmpT-F | GGAATTCCATATGCGGGCGAAACTTCTGGGAATAGTCCTGACAACCCCTATTGCGATCAGCTCTTTTGCTCCATGGGCTACATGGGTCACA |
| P16 | PhoA-F | GGAATTCCATATGAAACAAAGCACTATTGCACTGGCACTCTTACCGTTACTGTTTACCCCTGTGACAAAAGCCCCATGGGCTACATGGGTCACA |
| P17 | PelB-F | GGAATTCCATATGAAATACCTGCTGCCGACCGCTGCTGCTGGTCTGCTGCTCCTCGCTGCCCAGCCGGCGATGGCCATGGGCTACATGGGTCACA |
| P18 | General reverse primer | CTCGAGCG+GAGCGTTGCAGATACCAC |

PLD *: the PLD$_{PMF}$ with the native signal peptide (Nat) removed.

For secretory expression of PLD, $PLD_{PMF}$ was cloned into plasmid pET22b together with different signal peptides. Primers P5/P6 were used for PCR amplification of the $PLD_{PMF}$ gene containing the native signal peptide (Nat), while the primers P7/P8 were used to obtain the fragment OmpA-Nat-PLD *, and primers P9/P10 were used to obtain the fragment OmpA-PLD *. These three fragments were inserted into *Nco*I/*Xho*I sites of plasmid pET22b to yield the plasmids pET22b-Nat-PLD *, pET22b-OmpA-Nat-PLD *, and pET22b-OmpA-PLD * respectively. To optimize the secretory efficiency, seven other signal peptides OmpC, OmpF, OmpT, LamB, PhoA, MalE and PelB were amplified using appropriate primers listed in Table 3 to replace OmpA of plasmid pET22b-OmpA-Nat-PLD *.

### 4.3. Protein Expression and Cell Fractionation

The engineered *E. coli* was cultivated in 100 mL LB medium with 0.1 mm of appropriate antibiotics at 37 °C on a rotatory shaker (200 rpm). When the cell-culture density at 600 nm ($OD_{600}$) reached 0.6, 0.5 mm β-d-1-thiogalactopyranoside (IPTG) was added into the culture. Then the cells were incubated at 28 °C for 12 h. As shown in Figure 7a, cells were harvested by centrifugation at 8000 rpm for 10 min. The supernatant was used as the extracellular fraction to test the activity of extracellular PLD secreted into the culture medium. The collected cells were washed twice with deionized water and was resuspended in water to an $OD_{600nm}$ of 20 for the preparation of intracellular fraction. Intracellular fraction was prepared on ice by ultrasonication: 20 min pulsing (0.3 ms, 0.2 ms pause) at 40% input power and insoluble fraction of the lysate was removed by centrifugation (8000 rpm for 30 min at 4 °C). The intracellular fraction was used to evaluate the intracellular PLD activity.

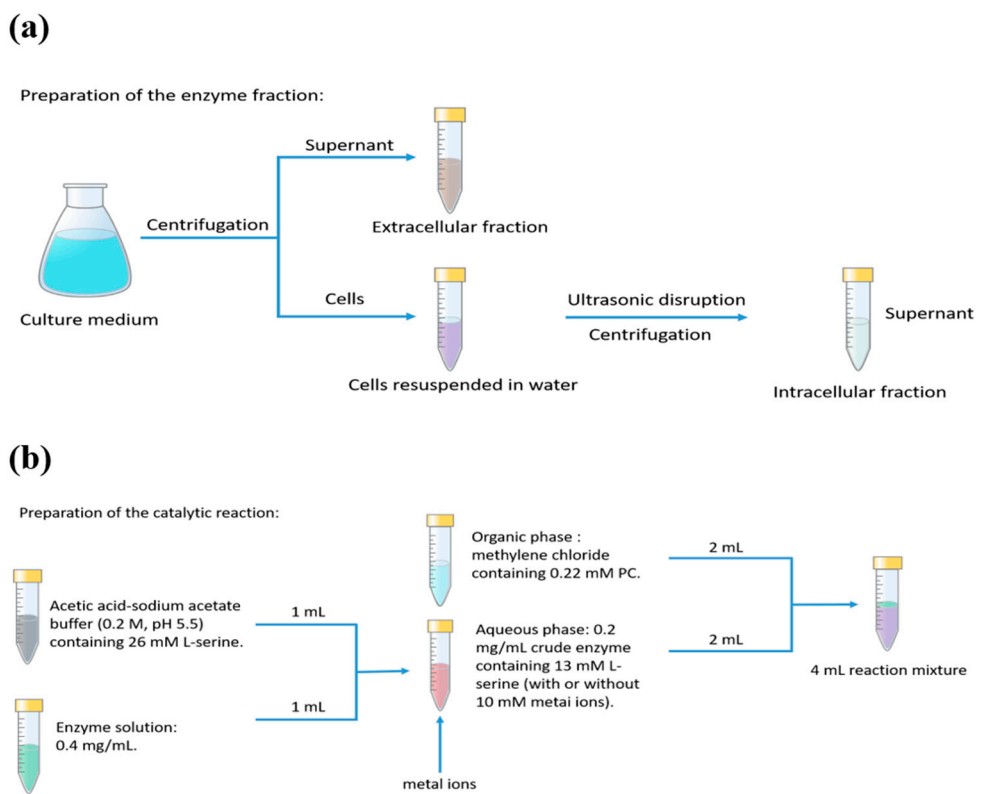

**Figure 7.** Flow charts of crude enzyme preparation and catalytic reaction. (**a**) Preparation of the crude enzyme. (**b**) Preparation of the catalytic reaction.

### 4.4. Enzyme Assay

The transphosphatidylation activity of PLD was measured according to the production of PS. One unit (U) was defined as 1 μmol PS produced per 1 min. To determine the PLD activity, a catalytic reaction was carried out in a two-phase system (Figure 7b). The aqueous phase with or without 10 mm

metal ions ($Co^{2+}$, $Ni^{2+}$, $Zn^{2+}$, $Cu^{2+}$, $Ca^{2+}$, $K^+$, $Fe^{2+}$, $Mg^{2+}$, $Mn^{2+}$) containing 13 mm L-serine consists of 1 mL enzyme solution (intracellular fraction or extracellular fraction containing 0.4 mg crude enzymes) and 1 mL acetic acid–sodium acetate buffer (0.2 M, pH 5.5). The organic phase was 2 mL methylene chloride containing 0.22 mm PC. The reaction mixture was incubated in a 200 rpm shaker at 30 °C for 4 h. Then, the reaction mixture was centrifuged (8000 rpm, 4 °C, 10 min) and the organic phase was retained. After that, a 100 μL organic sample was taken from the methylene chloride solution and diluted 10 times with a mixture containing chloroform and methanol with a volume ratio of 2:1. Then, the diluted sample was determined by HPLC.

*4.5. Analytical Methods*

The samples were analyzed by high-performance liquid chromatography (HPLC) (Agilent 1260, Palo Alto, California, USA) equipped with a CHROMA-CHEM evaporative light scattering detector (ELSD). The separation of PS and PC was performed on a ZORBAX Rx-SIL silica gel column (5 μm, 250 mm × 4.6 mm, Agilent). Mobile phase A contained 85% methanol, 14.5% water, 0.45% acetic acid and 0.05% trimethylamine; mobile phase B contained 20% n-hexane, 48% isopropanol and 32% mobile phase A. The flow rate was 1.0 mL min$^{-1}$. The elution conditions were as follows: initially mobile phase was 2% A and 98% B; 10% A and 90% B elute for 5 min; 30% A and 70% B elute for 4 min; 10% A and 90% B elute for 5 min; finally, 2% A and 98% B elute for 3 min. The column temperature, nebulizing temperature and evaporating temperature were controlled at 38 °C, 72 °C and 72 °C, respectively, and nitrogen was used as the nebulizing gas. The nitrogen gas flow rate was 2.0 SLPM (standard liters per minute). Each phospholipid was determined from the retention time using calibration solutions of corresponding phospholipids, and the concentrations of the phospholipids in the samples were calculated from the peak areas by integration.

## 5. Conclusions

In this study we cloned and expressed the PLD from *Streptomyces* PMF in *E. coli*; the strain BL21(DE3)/pET28a-PLD$_{PMF}$ only exhibited intracellular PLD activity. In order to release the negative effects caused by the toxicity of PLD we constructed the strain BL21 (DE3)/pET22b-PLD$_{PMF}$ to secrete the PLD into the culture medium and the supernatant of the culture exhibited PLD activity producing 1.23 g/L PS in 8 h. Then, we investigated the effects of signal peptides and adding surfactants on the secretory production of PLD. Strain BL21(DE3)/PelB-Nat-PLD * showed the highest extracellular PLD activity. With the addition of 3 g/L Span60, the extracellular fraction was used for catalytic reaction, and the concentration of PS reached 4.51 g/L, 3.67-fold higher than strain BL21 (DE3)/pET22b-PLD$_{PMF}$. After optimizing the induction conditions and catalytic situations, the recombinant PLD of the strain BL21 (DE3)/PelB-Nat-PLD * produced 30.2 g/L PS in 24 h. With the advantages of simple operations, low cost of recycling the PLD and high activity of the enzyme, our work makes large-scale production of PLD and PS feasible.

**Author Contributions:** Conceptualization, X.W., K.C. and P.O.; data curation, J.W., S.X. and X.W.; formal analysis, S.X.; funding acquisition, X.W. and P.O.; investigation, J.W. and Y.P.; methodology, S.X. and Y.P.; project administration, K.C.; writing—original draft, J.W.; writing—review and editing, J.W. and X.W. All authors have read and agreed to the published version of the manuscript.

**Funding:** This research was funded by the National Key Research and Development Program of China (Grant No. 2018YFA0901500), National Natural Science Foundation of China (Grant No. 21606127, Grant No. 21706126) and, Jiangsu synergetic innovation center for advanced bio-manufacture (Grant No. XTB1802, Grant No. XTE1844).

**Conflicts of Interest:** The authors declare no conflict of interest.

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
