# Peer review of "Highly Efficient Extracellular Production of Recombinant Streptomyces PMF Phospholipase D in Escherichia coli"

_catalysts, doi:10.3390/catal10091057_

Round 1

Reviewer 1 Report

The article “High efficient extracellular production of recombinant Streptomyces PMF phospholipase D in Escherichia coli” in my opinion can be considered for publication after major revision. I would suggest a careful revision of the citations in the text and the addition of some experiments.

Introduction:

  • Line 50, the authors cited Carlo et al. The correct citation would be: Zambonelli et al. “Zambonelli, C. Cloning and expression in Escherichia coli of the gene encoding Streptomyces PMF PLD, a phospholipase D with high transphosphatidylation activity. Enzyme and Microbial Technology 2003, 33, 676-688, doi:10.1016/s0141-0229(03)00190-x.

Results:

  • The key point of the article is the optimization of the extracellular production of PLD, thanks to the use of the signal peptide, and its extracellular activity optimization. In my opinion some experiments are missing:
  • A comparison of intracellular and extracellular hydrolytic activity of native and recombinant PLD.
  • A western blot showing the intracellular and extracellular fraction of the native PLD.
  • A western blot to compare the extracellular levels of recombinant PLD with different signal peptides.

Reviewer 2 Report

This is a good manuscript which describes in a complete manner the expression of PLD enzyme in two Streptomyces, the improvement of the expression system and application to the synthesis of PS.  Modifications needed are in the following:

  1. the sentence in line 37 is confusing While it is clear that natural secretion is an important aspect, "high transphosphatidylation activity of native PLD" must be better connected in its significance
  2. line 48 put a space
  3. line 49 expressed in
  4. par 2.3 seems to be incomplete, please revise starting from last sentence and generalize according to experiments reported
  5. par 2.4 must be more complete, please discuss how boiling for enzyme inactivation is arranged in presence of methylene chloride
  6. par 2.5 please complete with more details about eluition times
  7. Introduce the result section by clearly describing in a schematic figure the enzymatic reaction
  8. after the first experiment clearly express on which strain all the work is continued
  9. for the experiments of Figure 4 parallelized details in experimental parts must be present, in particular issues (if any) related to the use of detergents in the double phase designed experiments must be discussed
  10. in figure 5 the relative activity must be specified to which conditions is referenced (which is 100%?)
  11. par 3.5 starts with a vague sentence that must be precise. Details of exmperimental part must be present in the corrispective section

Reviewer 3 Report

Dear Editor

I have read with interest the manuscript presented by Dr. Wang and collaborators, on the bio-production of phospholipase D (PLD) in E. coli.

The study consists in reporting optimised conditions for the extracellular production of a specific PLD (from Streptomyces PMF), following the usage of exporting signals. The resulting PLD resulted functionally active in the transformation of lecithin into diacylphosphoserine.

The study appears well done but the Authors are not particularly clear in their writing, I suggest some textual improvements for a best manuscript.

Here my comments to improve the current manuscript:

  • 1) lines 35-39: The authors have justified their choice of targeting Streptomyces PLD because of high transphosphatidylation activity and natural PLD secretion activity. This is acceptable, but more comments are required on other possible choices. In particular, is the transphosphatidylation activity the most relevant (i.e., industrial or academic) activity of PLD? Authors interested in other activities would find useful here a very short comment on other possibilities. And, in addition, are PLD from other sources good as well to produce PS or PG or other phospholipids? Is there any specificity/selectivity?
  • 2) Lines 146-148. Although I understand that the first experiment was just to evaluate which PLD from two different strain would be “best” produced in the E. coli system, the authors should remark and explain that comparing the crude extracellular products in terms of PC-to-PS conversion gives a result that is the combination of several factors, from efficient intracellular production, to export, to intrinsic enzyme catalytic activity. Therefore, selecting the Streptomyces PMF strain actually results from a combination of factors. From this study, it is impossible to determine whether the other strain “SR” would have performed better in any one of the individual factor that – in combination – led to a superior activity of PMF. Please comment on this aspect in the manuscript, highlighting that this first selection is somehow based on a combination of factors, and that the PLD from SR strain would in principle work as well as the PMF one, if individual factors were studied and optimised. This is particularly relevant also because the difference (Fig. 1) is not huge.
  • 3) Line 154: do the Authors mean, with “co-expressed” the production of two different proteic structures or a fused structure (1 chain)?
  • 4) Line 160-161: here the text is confusing, I was expecting a comment on the third case: OmpA-Nat-PLD*, but the Authors instead discuss apparently another experiment (not shown). Please explain.
  • 5) In the same part, while it is clear that 45.72% > 40.68% shown above (by the way, the significance of the 0.01 digits is questionable in my opinion), it is not clear if the authors refer to some other comparison. In fact, they comment on the difference between PLD* and PLDPMF in line 161. The entire paragraph is not clearly written and it should be re-written in more ordered manner, to parallel the data presented in figures. And especially explaining better what is the protein being expressed in each step. Initially, it should be said that the PLD already contained the Nat sequence (if I understood well). The entire section need some text revision and Authors should explain better from the beginning the structure of PLD including fused signal peptides in early experiments.
  • 6) Again, line 174 and following, I understand that these are fused sequences for exporting, not co-expressed proteins, and thus it is relevant to discuss here whether or not monitoring the PC-to-PS activity is really the best indicator of exporting. Actually, if these are fusion structures, the intracellular production could differ, in principle, so that the extracellular activity measures both the export capacity and the number of active proteins.
  • 7) Line 222: optimal pH is 5.5 not 6.5
  • 8) Do the Authors have an idea on the composition of the extracellular medium? How many proteins are there? Would it be possible to easily separate PLD from others? Did the Author perform a gel electrophoresis analysis?

Minor

  • 1) On line 50, please note that “Carlo” is the first name of the author of ref. 15. Zambonelli is the family name. Also, ref. 15 includes only the first author, while the publication has several authors.
  • 2) Line 111, please provide also a measure (or an estimate) of the sonicator input power
  • 3) Line 117: specify what is the concentration (or any other indication about the amount) of PLD in the aqueous phase, and indicate also composition of the aqueous phase including buffer type and metal ions (and refer to section 3.4 for more details).
  • 4) Section 2.5: which lipids were exactly used? In the material section is not specified, but in HPLC section a calibration line is evidently used. But what are the compounds for calibration?
  • 5) Line 204: the text is misleading: please specify that it is not the PLD activity that reach the maximum at pH 6.5 (because, I guess, all PC-to-PS experiments were carried out at the same pH), but this refers to the pH at which the bacteria were cultivated.

Round 2

Reviewer 2 Report

Authors changed and added informations as indicated and I am happy to see the quality of this manuscript increeased to reach its publicability. As I noticed still some minor flaw it can be changed editorially before publication, please notice line 113 the use of capital letter for Cell and possibly in other few places, therefore editorial spell check required for this.

This manuscript is a resubmission of an earlier submission. The following is a list of the peer review reports and author responses from that submission.